# High-performance hysteresis-free perovskite transistors through anion engineering

Huihui Zhu [1], Ao Liu [1], Kyu In Shim[2], Haksoon Jung[1], Taoyu Zou[1], Youjin Reo[1], Hyunjun Kim[1], Jeong Woo Han [2], Yimu Chen[3], Hye Yong Chu[4], Jun Hyung Lim[4], Hyung-Jun Kim[4], Sai Bai [5,6✉] & Yong-Young Noh [1✉]

Despite the impressive development of metal halide perovskites in diverse optoelectronics, progress on high-performance transistors employing state-of-the-art perovskite channels has been limited due to ion migration and large organic spacer isolation. Herein, we report high-performance hysteresis-free p-channel perovskite thin-film transistors (TFTs) based on methylammonium tin iodide ($MASnI_3$) and rationalise the effects of halide (I/Br/Cl) anion engineering on film quality improvement and tin/iodine vacancy suppression, realising high hole mobilities of 20 cm$^2$ V$^{-1}$ s$^{-1}$, current on/off ratios exceeding 10$^7$, and threshold voltages of 0 V along with high operational stabilities and reproducibilities. We reveal ion migration has a negligible contribution to the hysteresis of Sn-based perovskite TFTs; instead, minority carrier trapping is the primary cause. Finally, we integrate the perovskite TFTs with commercialised n-channel indium gallium zinc oxide TFTs on a single chip to construct high-gain complementary inverters, facilitating the development of halide perovskite semiconductors for printable electronics and circuits.

[1] Department of Chemical Engineering, Pohang University of Science and Technology, 77 Cheongam-Ro, Nam-Gu, Pohang 37673, Republic of Korea. [2] Department of Chemical Engineering and School of Interdisciplinary Bioscience and Bioengineering, Pohang University of Science and Technology, 77 Cheongam-Ro, Nam-Gu, Pohang 37673, Republic of Korea. [3] Ministry of Industry and Information Technology Key Lab of Micro-Nano Optoelectronic Information System, Harbin Institute of Technology, Shenzhen 518055, China. [4] R&D Center, Samsung Display Inc., Yongin 17113, Republic of Korea. [5] Institute of Fundamental and Frontier Sciences, University of Electronic Science and Technology of China, Chengdu 611731, China. [6] Department of Physics, Chemistry and Biology (IFM), Linköping University, Linköping SE-58183, Sweden. ✉email: sai.bai@liu.se; yynoh@postech.ac.kr

As an emerging class of semiconductors with remarkable properties, metal halide perovskites have enabled unprecedented performance improvements in diverse optoelectronic devices, such as solar cells[1–3], light-emitting diodes (LEDs)[4–7], and photo/X-ray detectors[8,9]. In comparison, their prospects in high-performance transistors, which are fundamental building blocks for modern electronics, remain to be seen[10,11]. The dominant lead halide perovskites in state-of-the-art optoelectronic devices, e.g. methylammonium lead iodide (MAPbI$_3$), exhibit small carrier effective masses comparable with silicon; however, their room-temperature charge-carrier mobilities are strongly limited by Fröhlich interactions due to the polar nature of Pb–halide bonds[12,13]. Furthermore, the severe ion migration induced by halide vacancies results in poor field-effect modulation and large current–voltage hysteresis[14–18]. After considerable efforts in ionic defect cleaning/healing, a MAPbI$_3$ transistor with a room-temperature field-effect mobility ($\mu_{FE}$) of ~4 cm$^2$ V$^{-1}$ s$^{-1}$ was recently demonstrated[19]. In comparison, tin halide perovskites are predicated to possess higher room-temperature mobilities than their Pb-halide analogues owing to the reduced Fröhlich effect[12]. However, current Sn-perovskite transistors rely on two-dimensional (2D) layered perovskites or 2D/three-dimensional (3D) hybrids, in which carrier transport can be hindered by the bulky/insulating organic spacers, posing a critical barrier for further performance enhancement[20–26].

Here, we demonstrate high-performance and hysteresis-free p-channel perovskite thin-film transistors (TFTs) with a 3D methylammonium tin iodide (MASnI$_3$)-based channel layer via rational halide anion (I/Br/Cl) engineering. The co-substitution of small amounts of bromide and chloride for iodide enhances the film quality and vacancy passivation, enabling TFTs with excellent electrical characteristics, such as a high $\mu_{FE}$ of ~20 cm$^2$ V$^{-1}$ s$^{-1}$, an on/off current ratio ($I_{on}/I_{off}$) of over 10$^7$, and a threshold voltage ($V_{TH}$) of 0 V. Unlike the dominant role of ion migration in causing the hysteresis of Pb-halide perovskite TFTs, we declare that the device hysteresis in Sn-based perovskite TFTs originates from minority carrier trapping at iodide vacancy sites. These deep traps are substantially reduced by proper Br and Cl co-substitution, which eliminates the hysteresis and provides high operational stability and reproducibility. By integrating the perovskite TFTs with n-channel indium gallium zinc oxide (IGZO) TFTs, we realised complementary inverters with high gain of 140 and noise margin of over 70%, suggesting great processability and compatibility for large-area electronic circuits.

## Results

**MASnX$_3$ TFT performance.** Perovskite films were spin-coated on hafnium (IV) oxide (HfO$_2$) layers fabricated by atomic layer deposition (ALD) from precursors consisting of methylammonium iodide (MAI) and tin(II) halide (SnX$_2$, X = Cl, Br, I), and then thermally annealed (see more details in Methods). Note that small amount of SnF$_2$ addition in precursors is needed to get field-effect current modulation owing to the hole-suppression effect (Supplementary Fig. 1). Subsequently, gold source/drain electrodes were deposited, constructing the bottom-gate, top-contact TFTs (Fig. 1a). The TFTs obtained without halide engineering, denoted by 'I-pristine', exhibited typical p-channel transfer characteristics under continuous mode at room temperature. The representative I-pristine device exhibited a low gate leakage current of ~10$^{-10}$ A, large $I_{on}/I_{off}$ ratio exceeding 10$^6$, and maximum $\mu_{FE}$ of 1.3 cm$^2$ V$^{-1}$ s$^{-1}$ (Fig. 1b). This is the first demonstration of p-channel TFTs based on 3D MASnI$_3$ perovskite films, demonstrating electrical parameters comparable to those of previously reported perovskite TFTs (Supplementary Table 1).

We achieved TFTs with much improved performance by carefully engineering the halide compositions of the precursors for perovskite film deposition (Fig. 1b and Supplementary Fig. 2), inspired by the fact that recent breakthroughs in perovskite photovoltaics have been made mostly based on multiple compositions[3,27–29]. Specifically, partially substitution of the iodide source with bromide salt (2 mol%, the devices are denoted by 'I/Br') resulted in an over three-fold improvement in the $\mu_{FE}$ of the TFTs (4.3 cm$^2$ V$^{-1}$ s$^{-1}$). With the channel films deposited from a precursor with 6 mol% chloride substitution, we obtained devices (denoted by 'I/Cl') exhibiting an even higher $\mu_{FE}$ (9.5 cm$^2$ V$^{-1}$ s$^{-1}$). Surprisingly, a rational combination of the two halide engineering strategies, i.e. employing channels deposited from perovskite precursors with simultaneous Br and Cl substitution (2 mol% Br and 6 mol% Cl), led to greatly enhanced TFT (denoted by 'I/Br/Cl') performance. As shown in Fig. 1b, the optimised I/Br/Cl perovskite TFTs exhibited a $\mu_{FE}$ of 19.6 cm$^2$ V$^{-1}$ s$^{-1}$ with an $I_{on}/I_{off}$ of 3 × 10$^7$, which is superior to reported Pb- and 2D Sn-based perovskite TFTs (Supplementary Table 1). Textbook-like output curves ($I_{DS}$ versus $V_{DS}$) with clear linear and saturation currents were observed for all the devices (Supplementary Fig. 3), indicating an Ohmic contact between the channel films and electrodes and validating the reliability of mobility extraction[30]. Furthermore, the I/Br/Cl device operated in an ideal enhancement mode with a $V_{TH}$ of 0 V (Supplementary Fig. 4), suggesting that no applied bias voltage is needed to turn off the transistor, which is highly desirable for simplifying circuit design and minimising power consumption in practical applications[31].

In addition to higher mobilities than those of I-pristine, I/Br, and I/Cl devices, the TFTs based on I/Br/Cl perovskite channels also exhibited significantly reduced, even negligible, current–voltage hysteresis. To quantitatively analyse the hysteresis for the I-pristine, I/Br, I/Cl, and I/Br/Cl TFTs, we calculated the difference in $V_{GS}$ ($\Delta V_{GS}$) at $|I_{DS}| = 10^{-7}$ A, halfway between the on and off states[32], and presented the data in Fig. 1c. Notably, the I/Br/Cl devices exhibited an average $\Delta V_{GS}$ of 0.1 V, which is less than 1/10 of that of the other three types of TFTs, in which ion migration and/or carrier trapping probably occurred (discussed later). The negligible hysteresis for the I/Br/Cl devices is comparable with commercialised amorphous metal oxide TFTs[33]. Similar to previous perovskite solar cells[34], the hysteresis in the dual-sweep transfer curves of the TFTs also causes variations of the extracted performance parameters. We observed notable differences in the maximum mobility values extracted from the reverse (on-to-off, −12 to 7 V) and forward (off-to-on, 7 to −12 V) scans of the I-pristine, I/Br, and I/Cl TFTs (Supplementary Fig. 5), and presented the $\mu_{FE}$ statistics in Fig. 1d. The I/Cl TFTs demonstrated the largest mobility variations (>70%) because of their largest hysteresis, while the variations for the I-pristine and I/Br devices were slightly lower (50%). Notably, the I/Br/Cl devices exhibited the smallest mobility variations (12%) owing to their greatly reduced hysteresis. Because a strong mobility–hysteresis correlation exists but has been usually neglected in previous research on perovskite TFTs[17], we recommend more information on the measurement methods, device hysteresis, and mobilities extracted from both the forward and reverse scans of devices be provided. The hysteresis-free character is desired for a wide range of electronic applications, such as logic circuits and backplanes in OLED displays[35].

**Channel film characterisations.** To comprehensively understand the benefits of halide engineering for perovskite TFTs, we performed a series of film characterisations. The scanning electron microscope (SEM) images in Fig. 2a reveal a few pinholes in the I-pristine perovskite film, and slight Br substitution suppressed the

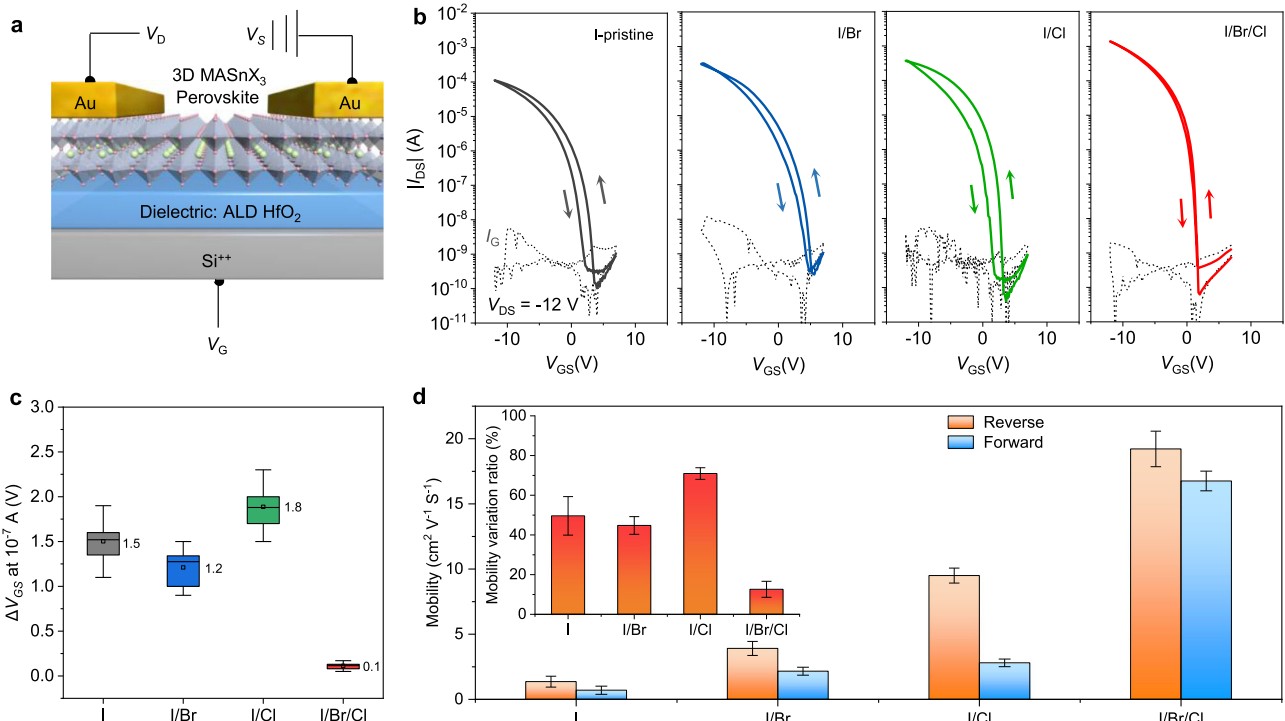

**Fig. 1 Electrical characteristics of MASnX₃ perovskite TFTs. a** TFT structure used in this work. **b** Transfer characteristics of the TFTs with different perovskite channel layers. $I_G$: gate leakage current. **c** Hysteresis statistics of different TFTs. The error bars present standard errors calculated from ten devices per type, and the mean values are labelled. **d** Histogram of the extracted mobilities from the transfer characteristics under different scan directions. The error bars present standard errors calculated from ten devices per type. The inset shows the variation ratio of the mobility values extracted from reverse ($\mu_{Rev}$) and forward ($\mu_{For}$) scans, calculated by ($\mu_{Rev} - \mu_{For})/\mu_{Rev} \times 100\%$.

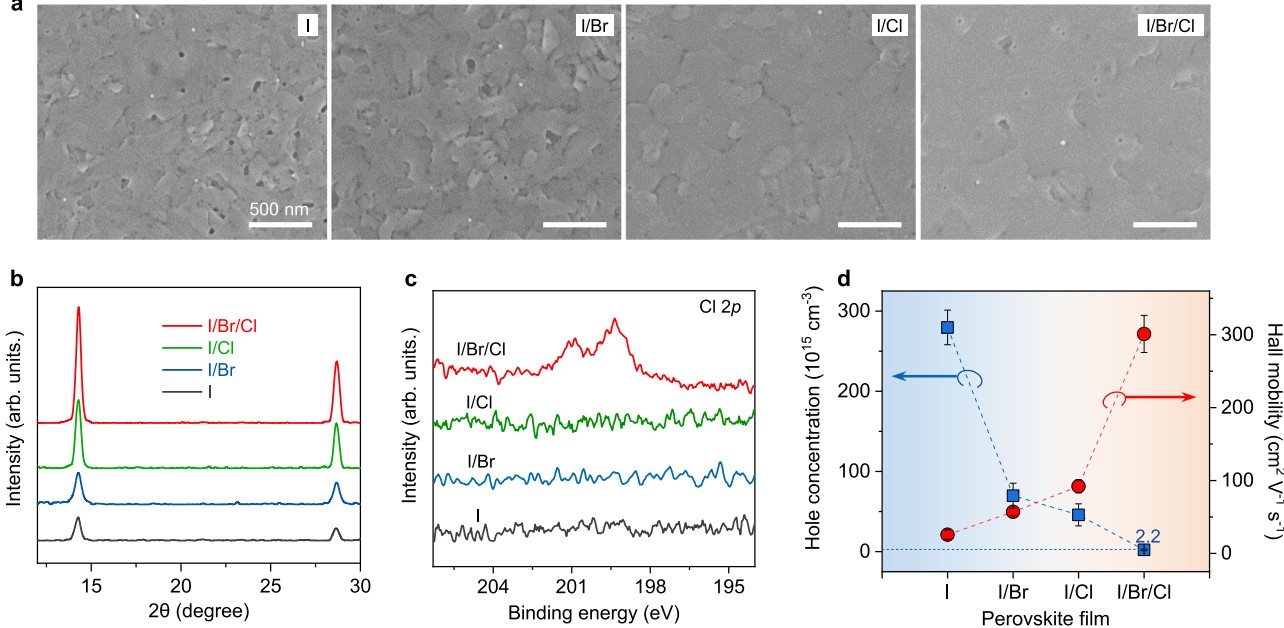

**Fig. 2 Characterisation of MASnX₃ films. a** SEM images, **b** XRD patterns, **c** Cl 2p core level XPS spectra, and **d** Hall mobilities and hole concentrations of the different perovskite films. The error bars present standard errors calculated from five films per type.

pinholes. The incorporated Br anions could compete with I anions and coordinate more strongly with the metal ions ($Sn^{2+}$), modulating the nucleation and crystallisation kinetics of the perovskite films[36,37]. In comparison, both the I/Cl and I/Br/Cl films exhibited a considerably smoother surface morphology,

suggesting the incorporation of Cl in the precursor significantly promoted perovskite formation, similar to observations about Cl in lead halide perovskites for solar cells[36,38,39]. The results are consistent with X-ray diffraction (XRD) analyses, where the I/Cl and I/Br/Cl samples exhibited substantially increased intensities

of the main diffraction peaks compared with those of the I-pristine and I/Br samples, suggesting improved crystallinity and/or improved grain orientation (Fig. 2b).

Interestingly, both the I/Br and I/Br/Cl films exhibited identifiable XRD peak shifts to higher angles compared with that of the I-pristine film. The peak shifts indicate a reduced $d$-spacing due to the incorporation of smaller Br and/or Cl ions into the I-based perovskite lattices. However, the I/Cl sample showed negligible peak shifts (Supplementary Fig. 6), which suggests that the Cl anions in the I/Cl perovskite film did not enter the perovskite lattice but only functioned to improve the film morphology and crystallinity[40]. The X-ray photoelectron spectroscopy (XPS) Cl $2p$ core level spectra further confirmed that Cl was undetectable in the I/Cl films but was successfully incorporated into the triple-halide I/Br/Cl films (Fig. 2c). These results are unsurprising considering the large discrepancy in ionic size between I and Cl anions and the potential volatilisation of Cl additives during film annealing[41]. However, in the I/Br/Cl perovskite film, the Br-substituted I-based lattice was capable of hosting Cl anions owing to the bridging effect of Br[42,43], enabling the formation of the triple-halide MASn(I/Br/Cl)$_3$ perovskite. We estimated the relative atomic concentration ratio of I:Br:Cl from the XPS data for the triple-halide sample. The relative ratio is 93.8%:2.4%:3.8% and the feeding ratio was 92%:2%:6%, consistent with above discussion.

We then conducted Hall-effect measurements to investigate the carrier concentrations and Hall mobilities of the perovskite films, which are important for understanding the performance of the resulting TFTs. As shown in Fig. 2d, the I-pristine films exhibited an average hole concentration of $2.8 \times 10^{17}$ cm$^{-3}$, which gradually decreased to $6.9 \times 10^{16}$, $4.5 \times 10^{16}$, and $2.2 \times 10^{15}$ cm$^{-3}$ for the I/Br, I/Cl, and I/Br/Cl films, respectively. This trend resulted from decreasing hole sources (tin vacancies). Generally, in 3D Sn-based perovskites, Sn$^{2+}$ can easily oxidise into Sn$^{4+}$ even under trace oxygen, and intrinsic tin vacancies ($V_{Sn}$) have a low formation energy, which both cause notoriously high hole concentrations. The Sn $3d_{5/2}$ XPS spectra (Supplementary Fig. 7) showed that the Sn$^{4+}$ signal gradually decreased in the sequence of I > I/Br > I/Cl > I/Br/Cl, indicating suppressed Sn$^{2+}$ oxidation. Furthermore, previous studies have suggested that the incorporation of anions with electronegativity stronger than that of I$^-$ raises the $V_{Sn}$ formation energy during perovskite crystallisation, further reducing the hole concentration[24,44]. Additionally, for the I-pristine sample, a small shoulder peak appeared at a lower binding energy (~485.6 eV), which is ascribed to under-coordinated Sn with an oxidation state of $\delta < 2^+$ (Sn$^{\delta < 2+}$)[45]. However, this shoulder peak was undetectable in other samples, indicating well-coordinated Sn sublattices and reduced structural imperfections with halide engineering.

The Hall mobilities ($\mu_{Hall}$) of the perovskite films showed a consistent trend with the $\mu_{FE}$ extracted from the corresponding TFTs, with the average $\mu_{Hall}$ increasing from 25 cm$^2$ V$^{-1}$ s$^{-1}$ (I-pristine) to 57 (I/Br) and 92 cm$^2$ V$^{-1}$ s$^{-1}$ (I/Cl), respectively. The $\mu_{Hall}$ for I/Br/Cl films was up to 301 cm$^2$ V$^{-1}$ s$^{-1}$. According to the simple Drude model, the hole mobility of a p-type semiconductor is determined by the hole effective mass ($m_h^*$) and the average scattering time ($\tau$):

$$\mu = q\tau/m_h^* \tag{1}$$

where $q$ is the elementary charge[12]. Considering the small halide substitution in the perovskite lattice, negligible changes to $m_h^*$ were expected. Therefore, $\mu_{Hall}$ should be mainly determined by the scattering time interval during carrier transport, which is dominated by scattering centres, e.g. ionised (negatively and positively charged) defects and crystal disorders, in the perovskite films. As revealed by the characterisations above, halide engineering effectively enhanced the film quality and reduced ionised

defects, particularly in the I/Br/Cl film, significantly suppressing charge-carrier scattering and providing a rationale for the enhanced $\mu_{Hall}$ of the halide-engineered perovskite films.

In addition to the improved perovskite film quality, the considerably improved performance of the I/Br/Cl devices is related to the properties of the dielectric–perovskite interfaces. Generally, the density of states ($N_s$) at the interface, which negatively affects the device performance of TFTs, can be estimated from the average subthreshold swing (SS, Supplementary Fig. 8):

$$SS = \frac{\kappa T ln 10}{e}\left[1 + \frac{e^2}{C_i}N_S^{max}\right] \tag{2}$$

where $\kappa$ is the Boltzmann constant, $e$ is the electron charge, and $C_i$ is the areal capacitance of the dielectric layer[46]. Accordingly, $N_S^{max}$ of the I-pristine TFTs was calculated to be $1.5 \times 10^{13}$ cm$^{-2}$ eV$^{-1}$, which reduced to ~$3.3 \times 10^{12}$ cm$^{-2}$ eV$^{-1}$ in the I/Br/Cl TFTs. This suggests that Br and Cl co-substitution enhanced not only the film quality but also the dielectric–perovskite interfaces of the TFTs.

## Understanding of the hysteresis

Having elucidated the benefits of rational halide engineering of MASnI$_3$ precursors on the TFT performance, we attempted to gain an in-depth understanding of the hysteresis behaviour of the devices. The commonly observed hysteresis of transistors utilising 3D lead-halide perovskite films is typically attributed to ion migration in the perovskite channel[47]. Such ion-migration-induced hysteresis is strongly dependent on the sweep rate during device measurement. For example, transistors based on single crystalline MAPbX$_3$ channels exhibited gradually expanded hysteresis when the sweep rate increased from 0.05 to 0.25 V s$^{-1}$(ref. [48]). However, we observed negligible changes to the hysteresis in the transfer curves of both the I-pristine and I/Br/Cl devices when the sweep rate increased from 0.4 to 4 V s$^{-1}$ (Fig. 3a), suggesting that the ion migration in the MASnI$_3$-based perovskite films did not contribute significantly to the TFT hysteresis. This can be partially explained by the different defect properties between Pb- and Sn-based perovskites[28]. In p-type MASnI$_3$, tin vacancies (electron acceptors) are the dominant defects, whereas the iodine defects, e.g. iodine vacancies ($V_I$) and interstitials ($I_i$), are much less (if even present)[49]. Consequently, the defect-associated migration of iodide ions, which have the lowest activation energy and move most easily, is less significant in MASnI$_3$ than in Pb-based perovskites, greatly reducing the associated electric-field screening effects during TFT operation.

We then considered charge-carrier trapping as the primary reason for the hysteresis in our perovskite TFTs, which exhibited a higher current in the transfer curves during the off-to-on sweep than that during the on-to-off sweep. The established models indicate deep electron and hole traps in the semiconductor channels are possible causes for this type of hysteresis in TFTs[50]. Theoretical calculations have predicted that in MASnI$_3$, hole traps induced by $I_i$ and $V_{Sn}$ defects are shallow, with thermodynamic ionisation levels close to or inside the valance band maximum[49]. Thus, they are expected to mainly affect the $\mu_{FE}$ and SS of the TFTs rather than induce hysteresis[50]. We postulated that $V_I$ defects, which possess the lowest formation energy among possible deep electron traps in MASnI$_3$, were the root cause of the hysteresis in the p-channel TFTs. As shown in the inset of Fig. 3a, when $V_{GS} \ll V_{TH}$, negative charge accumulated in the channel, and $V_I$-related long-lifetime electron traps were filled. The trapped electrons shifted the flat-band voltage; that is, the threshold voltage was reduced. When $V_{GS}$ swept towards negative potentials, more holes were induced, leading to a higher drain current.

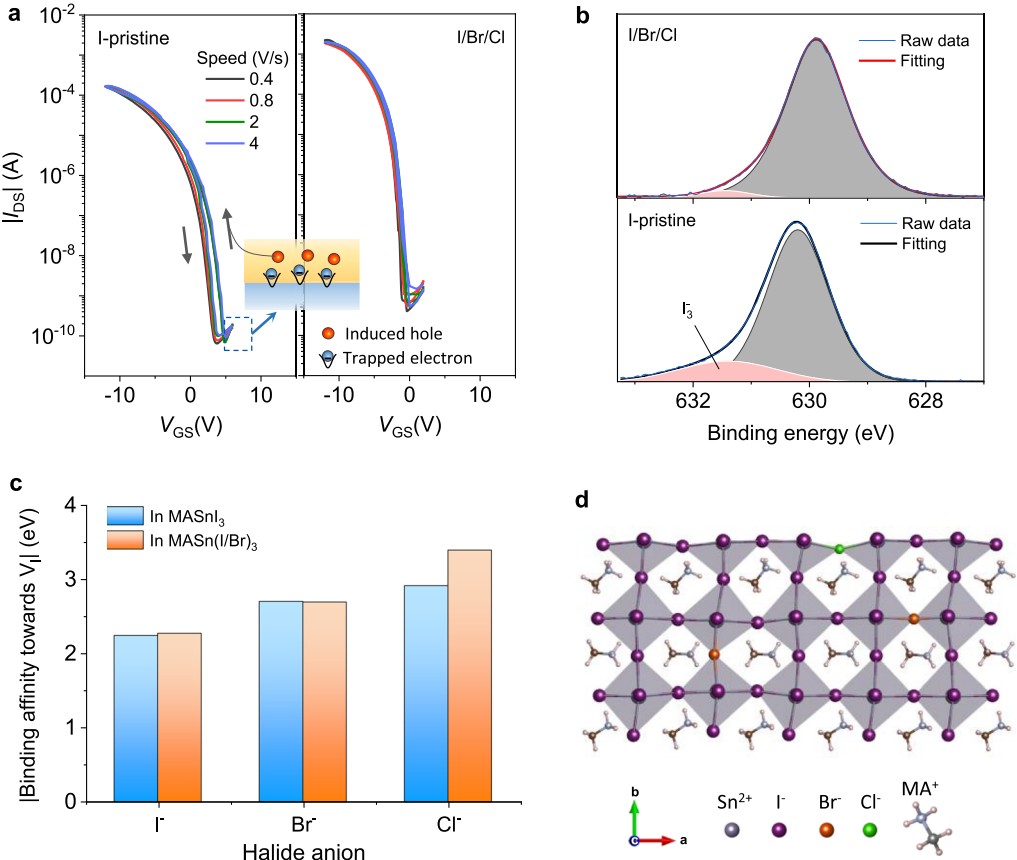

**Fig. 3 Hysteresis and $V_I$ in the MASnX$_3$ perovskite TFTs. a** Transfer characteristics of I-pristine and I/Br/Cl TFTs measured at different scan speeds. **b** I $3d_{3/2}$ core level spectra of the I-pristine and I/Br/Cl perovskites. **c** Calculated relative interaction strengths of halide anions with $V_I$ sites in MASnI$_3$ and MASn(I/Br)$_3$. **d** Illustration of the passivation effects of a $V_I$ defect by a Cl anion.

In comparison, the on-to-off sweep started directly from hole accumulation at $V_{GS} = -12$ V without the influence of stored negative charge[50], demonstrating a lower drain current. Therefore, the on-to-off measurement of the perovskite TFTs should more closely resemble the ideal field-effect transistor model, indicating the mobility extracted from the on-to-off transfer curve may be closer to the actual $\mu_{FE}$.

With the hypothesis that deep electron traps dominate the hysteresis in our p-channel perovskite TFTs, we investigated the different $V_I$ properties of the I-pristine and I/Br/Cl perovskite films to unveil the mechanism that eliminated hysteresis in the optimised I/Br/Cl devices. In the I $3d_{3/2}$ core level XPS spectra (Fig. 3b), the peak for the I-pristine sample shifted by 0.4 eV towards higher binding energies compared with that of the I/Br/Cl perovskite. This peak shift was previously ascribed to iodine loss from the lattice[51,52], indicating a higher probability of $V_I$ formation in the I-pristine perovskite film. In addition, a shoulder peak appeared at ~631.5 eV in the I $3d_{3/2}$ core level spectra of the I-pristine film, corresponding to the I$_3^-$ species. These species were assigned to iodide interstitials/$V_I^+$ iodine Frenkel defects, which form preferentially under $V_{Sn}$-rich conditions[53]. The shoulder peak became negligible for the I/Br/Cl perovskite, which can be attributed to the reduced iodine loss from the perovskite lattice, along with the significantly reduced hole concentration ($V_{Sn}$ defects) revealed by the Hall measurements. Density functional theory (DFT) calculations further confirmed the benefits of passivating $V_I$ sites in the I/Br/Cl perovskite film. As shown in Fig. 3c, Br or Cl anions, if successfully incorporated into the MASnI$_3$ perovskite lattice, possess higher binding affinities towards $V_I$ sites than that of I anions (slab models in Supplementary Fig. 9), in agreement with recent Pb perovskites

with double anions[3]. Based on the MASn(I/Br)$_3$ perovskite, the calculated binding affinity of a third anion, Cl$^-$, to $V_I$ was further enhanced (Figs. 3c, d), and hence the $V_I$ sites in the I/Br/Cl perovskite were expected to be greatly suppressed, rationalising the elimination of hysteresis in the resulting TFTs.

**TFT stability and complementary inverter.** We then characterised the operational stability of our perovskite TFTs, which is another critical figure of merit for practical applications. We first monitored the on/off switching stability of the devices (Fig. 4a). The I-pristine device exhibited an obvious current decay during the consecutive on/off switching test, while the currents of both the on and off states of the I/Br/Cl device remained consistent. We also examined the device stability under dynamic $V_{GS}$ scans. The transfer characteristics of the I-pristine TFTs gradually shifted, while those of the I/Br/Cl TFTs overlapped completely over 100 cyclic sweeps (Supplementary Fig. 10), suggesting considerably enhanced reliability of the I/Br/Cl devices. To evaluate the stability of the TFTs more rigorously, we performed a bias stress test, during which $-12$ V was applied constantly, and the shift in $V_{TH}$ was monitored. As shown in Fig. 4b, the $V_{TH}$ of the I-pristine device shifted significantly by $-2$ V (~17% of the operating voltage) after only 1000 s during the bias test (Supplementary Fig. 11), reflecting the serious carrier trapping in the devices[54]. Encouragingly, the optimised I/Br/Cl TFT exhibited much improved stability with a small threshold voltage shift ($\Delta V_{TH}$) of 0.52 V even after biasing for 12 h, approaching the stability of previously demonstrated stable transistors based on organic and amorphous silicon channels[47,55]. In addition, we

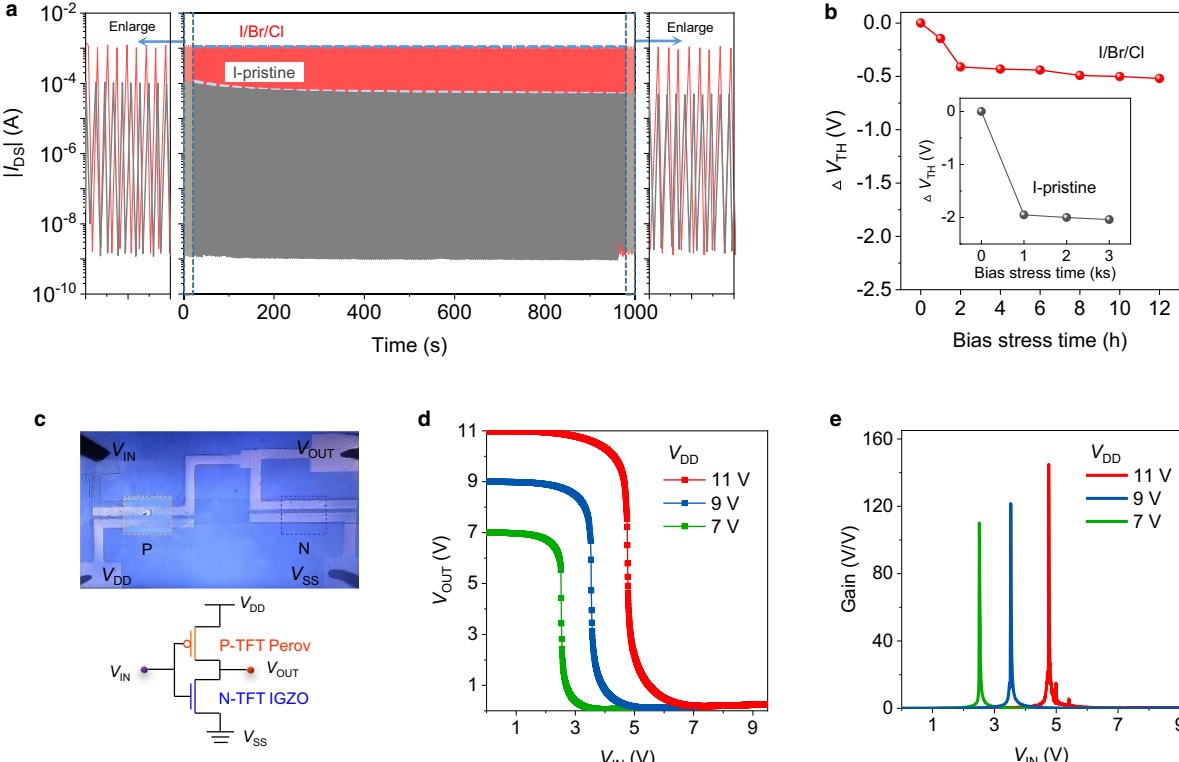

**Fig. 4 Operational stability of perovskite TFTs and performance of the integrated inverters. a** On/off switching sweep of the I-pristine and I/Br/Cl TFTs. **b** $V_{TH}$ variation under bias ($V_{GS} = V_{DS} = -12$ V). **c** Optical image and diagram of an integrated perovskite/IGZO inverter. **d** Voltage transfer characteristics and **e** gain (($dV_{OUT}$)/($dV_{IN}$)).

monitored the device air stability and found that the TFTs were unstable in air with fast device degradation due to the easy $Sn^{2+}$ oxidisation. However, without air exposure (stored in vacuum, ~$1 \times 10^{-6}$ Torr), the device showed high stability with constant transfer characteristics after a test period for 60 days (Supplementary Fig. 12). We anticipate that effective encapsulations, such as those used in protecting organic electronics, and/or anti-oxidation additives for Sn-based perovskites would be helpful to improve the air stability in the near future[56].

With the reproducible and stable p-channel I/Br/Cl perovskite TFTs (Supplementary Fig. 13), we moved a step further to explore their compatibility with existing TFTs based on n-channel metal oxides for monolithic complementary circuit integration. We fabricated complementary inverters by integrating perovskite TFTs with IGZO TFTs on a single chip. Figure 4c shows an optical image and diagram of the complementary inverter. The standard rail-to-rail voltage transfer characteristics of the inverter at different $V_{DD}$ (Fig. 4d) show $V_{OUT}$ being either 0 V or the supplied $V_{DD}$, suggesting an ideal logic '1' to '0' transfer. The inverter exhibited a high gain of over 100 at all measured $V_{DD}$ and a peak gain of 140 at $V_{DD} = 11$ V (Fig. 4e), which is significantly higher than that of the wire-connected complementary inverter or CMOS-like inverters involving perovskite TFTs in previous studies[24,57]. To the best of our knowledge, this is also the first demonstration of the monolithic integration of a complementary circuit involving perovskite TFTs. Additionally, the noise margin of the inverter was 3.93 V, reaching 72% of the ideal value ($V_{DD}/2$) (Supplementary Fig. 14), which is sufficient for most static logic applications[58].

In conclusion, we have achieved high-performance and hysteresis-free MASnI$_3$-based perovskite TFTs through rational halide engineering. We revealed the benefits of Br and Cl co-substitution in the precursor: enhanced film quality and reduced

vacancy defects of the perovskite films, which enabled exceptional performance of the resulting TFTs. Moreover, we found ion migration had a negligible contribution to the hysteresis in our p-channel perovskite TFTs based on MASnI$_3$ films. Alternatively, we correlated the hysteresis of our MASnI$_3$ TFT with deep electron traps induced by $V_I$ defects, which are notably reduced by rational Br and Cl co-substitution in the precursor. By combining our operationally stable p-channel perovskite TFTs with n-channel IGZO TFTs, we demonstrated monolithically integrated high-gain complementary inverters, suggesting high compatibility and processability for electronic applications.

## Methods

**Precursor and device preparation**. The precursor solutions were prepared by mixing MAX and SnX$_2$ (X = I, Cl, Br) in DMF/DMSO binary solvents at a volume ratio of 4/1. The ratio of different composition is expressed as MASn(I$_{1-y}$X$_y$)$_3$, X = Cl or Br, and y = 0~0.08. The precursor concentration was around 0.15 M. For example, to prepare the MASnI$_3$ precursor, 102 mg of MAI, 216 mg of SnI$_2$, 17 mg of PbI$_2$, and 8.4 mg of SnF$_2$ were dissolved in 4 mL of mixed solvent (3.2 mL of DMF and 0.8 mL of DMSO). The others were made similarly. Same amounts of SnF$_2$ (8 mol% with respect to the Sn source) and lead substitution (6 mol%) were added in each precursor as anti-oxidation and stabiliser additives. Each precursor was stir-heated at 60 °C for 2 h before use. Dielectric layers were the reliable 40-nm HfO$_2$ on Si substrates with capacitance of ~270 nF cm$^{-2}$ grown by the mature atomic layer deposition (ALD) in National Institute for Nanomaterials Technology, Pohang. For the perovskite film deposition, the HfO$_2$/Si substrates ($1.5 \times 1.5$ cm$^2$) were first treated with UV ozone for 30 min, then the precursor solutions were dropped on the substrates and spin-coated at the speed of 4000 rpm for 25 s with 60 uL of chlorobenzene dripping after ~10 s from the beginning. Afterwards, the films were annealed at 70 °C for 10 min, constructing thin films with similar thickness of ~$40 \pm 5$ nm. All precursor solutions and films were prepared in an N$_2$-filled glove box (O$_2$ and H$_2$O levels: 1–2 ppm). Then thermal evaporation and a shadow mask were used for the Au source/drain electrode deposition. The channel width and length of the TFTs are 1000 and 150 um, respectively. For the complementary inverter integration, we used design-patterned indium tin oxide (ITO, 10 Ω/□) glass by photolithography as the bottom gate, ALD HfO$_2$ as the dielectric layer, solution-processed self-patterned[59] triple-halide

perovskite as the p-channel and IGZO as the n-channel, and evaporated gold through a bespoke shadow mask as electrodes.

**Film and device characterisations**. The film XRD patterns were recorded using a Rigaku D/MAX 2600 V with Cu Kα (λ = 1.5406 Å) radiation. SEM images were obtained using a field-emission scanning electron microscope (Hitachi S4800). AFM height profile data were collected using an atomic force microscopy (Nanosurf Nanite AFM). XPS characterisations were conducted using a VersaProbe Scanning Microprobe under vacuum ($10^{-8}$ Torr). The Hall measurements were performed using the van der Pauw method, using a 0.51T magnet and a bespoke sample holder in an $N_2$-filled glove box at room temperature. The electrical signal during the Hall measurement was obtained using a Keithley 4200-SCS and probe station (MST-4000A, MS TECH, Korea). Transistor transfer characteristics were measured using a semiconductor parameter analyser (Keithley 4200-SCS) in an $N_2$-filled glove box at room temperature in continuous mode. The saturation TFT mobility was calculated as[60]

$$\mu_{sat} = \frac{2L}{WC_i}\left(\frac{\partial\sqrt{I_{DS}}}{\partial V_{GS}}\right)^2 \qquad (3)$$

where $L$, $W$, and $C_i$ are the channel length and width and dielectric areal capacitance, respectively.

## Data availability

All data needed to evaluate the conclusion in the paper are present in the paper and/or the Supplementary Materials. Additional data related to this paper can be available from the authors upon reasonable request.

## Code availability

The custom software used for data acquisition and data analysis is available from the corresponding author upon reasonable request.

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

## Acknowledgements

This study was supported by the Ministry of Science and ICT through the National Research Foundation, funded by the Korean government (2021R1A2C3005401, and 2020R1A4A1019455). This research was also supported by the Samsung Display Corporation. H.H. Z. thanks Won-Tae Park, Ji-Young Go, and Gi-Seong Ryu for their help with the metal oxide TFTs.

## Author contributions

H.H.Z. and Y.Y.N. conceived the study. H.H.Z. and A.L. performed the experiments. A.L., T.Z., Y.R., H.K., H.Y.C., J.H.L., H-J.K., and assisted with the characterisations and analyses. K.I.S. and J.W.H. performed the DFT calculations. H.J., T.Z., and A.L. assisted with the circuit mask design. H.H.Z., A.L., S.B., and Y.Y.N. wrote the manuscript. Y.C. assisted with manuscript revision. A.L. and S.B. contributed to the experimental design and data analysis. Y.Y.N. supervised the project. All the authors contributed to the discussion. All authors approved the final version of the manuscript.

## Funding

## Competing interests

The authors declare no competing interests.
