## [Peer review file · Nature Communications]

REVIEWER COMMENTS

Reviewer #1 (Remarks to the Author):

I read carefully the response letter based on previous comments, I think the authors have almost addressed my concerns. There is still one question:

For the R1Q3 about the XPS results, the authors argued that the perovskite is inevitably exposed to the air, I suggest the authors to coat some inert materials such as PMMA on perovskite and then carried out XPS measurement, after etching the PMMA in XPS instrument, you can get more real result about the perovskite.

Reviewer #2 (Remarks to the Author):

The authors have provided a clear and satisfactory response to the issues raised in my report. However, one issue requires further clarification. The authors now argue that the concentration of Sn⁴⁺ defects might be due to accidental oxygen exposure of the films and they have moved the XPS results to the SI. I think this is not very satisfactory given that the question of Sn⁴⁺ defects is an important and widely appreciated concern.

I would encourage the authors to perform their XPS experiments under conditions where the effect of oxygen exposure can be eliminated by an inert atmosphere transfer into the vacuum chamber (or at least minimized by investigating how the Sn⁴⁺ concentration varies when the films are exposed to air for different length of times before the transfer into UHV.)

Reviewer #3 (Remarks to the Author):

The authors, have taken the corections very seriously and have submitted a very good revised manuscript that in my opinion should be publish without any further need of corrections.

Reviewer #1 (Remarks to the Author):

I read carefully the response letter based on previous comments, I think the authors have almost addressed my concerns. There is still one question:

For the R1Q3 about the XPS results, the authors argued that the perovskite is inevitably exposed to the air, I suggest the authors to coat some inert materials such as PMMA on perovskite and then carried out XPS measurement, after etching the PMMA in XPS instrument, you can get more real result about the perovskite.

Reply: We would like to thank the reviewer for reviewing our manuscript and for positive comments.

Following the reviewer's suggestion, we re-conducted the XPS measurement under a minimized oxygen exposure condition using high-vacuum metal/rubber/metal containers (which can be high-vacuumed under an inert atmosphere) instead of the previous low-vacuum package (**Figure R1**) for the sample transfer process. In this way, we obtained more reliable data.

We updated the data in Figure S7 and Figure 3b, also shown below. The new data on Sn $3d_{5/2}$ XPS analysis shows the same relative Sn⁴⁺ content trend ($I > I/Br > I/Cl > I/Br/Cl$) with the original ones but lower absolute value for each film, supporting the previous discussion in the main text well. The relative Sn²⁺/Sn⁴⁺ ratios are found reasonable compared to previous reports on Sn-based perovskites.¹⁻⁴

We also thank the reviewer for suggesting the PMMA coating/etching method. The high-power ion etching can cause the degradation of the local bonding environment considering the vulnerable property of halide perovskites, which may influence the measurement reliability. Therefore, we chose the method above, and it turned out effective.

Figure R1. Previous low-vacuum package with sample tray and the newly-developed

high-vacuum metal/rubber/metal container. The new container can be first high-vacuumed under an inert atmosphere using the anti-chamber in an N₂-filled glove box, then vacuumed with a plastic bag for sample transfer.

Supplementary Fig. 7. Analyses of the Sn 3d_{5/2} core level XPS spectra with fitting results for Sn^{δ<2+}, Sn²⁺, and Sn⁴⁺.

Figure 3. b I 3d_{3/2} core level spectra of the I-pristine and I/Br/Cl perovskites.

Reviewer #2 (Remarks to the Author):

The authors have provided a clear and satisfactory response to the issues raised in my report. However, one issue requires further clarification. The authors now argue that the concentration of Sn^{4+} defects might be due to accidental oxygen exposure of the films, and they have moved the XPS results to the SI. I think this is not very satisfactory given that the question of Sn^{4+} defects is an important and widely appreciated concern.

I would encourage the authors to perform their XPS experiments under conditions where the effect of oxygen exposure can be eliminated by an inert atmosphere transfer into the vacuum chamber (or at least minimized by investigating how the Sn^{4+} concentration varies when the films are exposed to air for different length of times before the transfer into UHV.)

Reply: We thank the reviewer for reviewing our work again and for the positive comments on our Response Letter. We also thank the reviewer's helpful suggestions on XPS experiment.

Inspired by the reviewer's good suggestion, we re-conducted the XPS measurement under a minimized oxygen exposure condition using high-vacuum metal/rubber/metal containers (which can be vacuumed under an inert atmosphere) instead of the previous low-vacuum package (**Figure R1**) for sample transfer. In this way, we obtained more reliable data.

We updated the data in Figure S7 and Figure 3b, also shown below. The new data on $\text{Sn } 3d_{5/2}$ XPS analysis shows the same relative Sn^{4+} content trend ($I > I/\text{Br} > I/\text{Cl} > I/\text{Br}/\text{Cl}$) with the original ones but lower absolute value for each film, supporting the previous discussion in the main text well. The relative $\text{Sn}^{2+}/\text{Sn}^{4+}$ ratios are found reasonable compared to previous reports on Sn-based perovskites.¹⁻⁴

Figure R1. Previous low-vacuum package with sample tray and the newly-developed high-vacuum metal/rubber/metal container. The new container can be first high-vacuumed under an inert atmosphere using the anti-chamber in an N_2 -filled glove box, then vacuumed

with a plastic bag before transfer.

Supplementary Fig. 7. Analyses of the Sn $3d_{5/2}$ core level XPS spectra with fitting results for $\text{Sn}^{\delta < 2+}$, Sn^{2+} , and Sn^{4+} .

Figure 3. b I $3d_{3/2}$ core level spectra of the I-pristine and I/Br/Cl perovskites.

Reviewer #3 (Remarks to the Author):

The authors, have taken the corrections very seriously and have submitted a very good revised manuscript that in my opinion should be publish without any further need of corrections.

Reply: We thank the reviewer's careful evaluation and positive comments on our manuscript.

References

1. Ye, T. et al. Ambient-air-stable lead-free CsSnI₃ solar cells with greater than 7.5% efficiency. *J. Am. Chem. Soc.* **143**, 4319-4328 (2021).
2. Lin, R. et al. Monolithic all-perovskite tandem solar cells with 24.8% efficiency exploiting comproportionation to suppress Sn(ii) oxidation in precursor ink. *Nat. Energy* **4**, 864-873 (2019).
3. Gong, J. et al. Suppressed oxidation and photodarkening of hybrid tin iodide perovskite achieved with reductive organic small molecule. *ACS Appl. Energy Mater.* **4**, 4704-4710 (2021).
4. Yuan, F. et al. Color-pure red light-emitting diodes based on two-dimensional lead-free perovskites. *Sci. Adv.* **6**, eabb0253.

----- end of the response letter.

REVIEWERS' COMMENTS

Reviewer #1 (Remarks to the Author):

The authors have tried their best to address the Reviewers' suggestions and update the data, the paper can be accepted for publication now.

Reviewer #2 (Remarks to the Author):

The authors have addressed my concerns about the XPS measurements and the paper can now be accepted without further revision.